# 3D Co-Seismic Surface Displacements Measured by DInSAR and MAI of the 2017 Sarpol Zahab Earthquake, Mw7.3

**DOI:** 10.3390/ijerph19169831

**Published:** 2022-08-10

**Authors:** Randa Ali, Xiyong Wu, Qiang Chen, Basheer A. Elubid, Dafalla S. Dafalla, Muhammad Kamran, Abdelmottaleb A. Aldoud

**Affiliations:** 1Department of Geological Engineering, Faculty of Geosciences and Environmental Engineering, Southwest Jiaotong University, High-Tech Zone, Chengdu 611756, China; 2Faculty of Engineering, China University of Geosciences, 388 Lumo Road, Wuhan 430074, China

**Keywords:** earthquake, co-seismic, 3D deformation, DInSAR, Zagros Mountains Front Fault, multiple apertures interferometry

## Abstract

On 12 November 2017, an earthquake occurred in Sarpol Zahab city, located on the Iraq/Iran boundary, with a moment magnitude (Mw) of 7.3. Advanced Land Observing Satellite 2 (ALOS-2) L-band (23.6 cm wavelength) and C-band Sentinel-1A data (ascending and descending) were used to detect the co-seismic displacements maps caused by this earthquake. The ALOS-2 data was utilized to reconstruct the 3D co-seismic displacements maps, as well as estimate the fault-dip and slip distribution along the rupture. The results showed the maximum surface displacement in the north, east, and up directions to be 100, 50, and 100 cm, respectively. The best-fit faulting geometry had a strike of 337.5° and a dip of 11.2° toward the northeast, at a depth of 8 km. The predicted geodetic moment was 1.15 1020 Nm, which corresponds to a magnitude of Mw 7.31. There were two significant slip sources: one in the shallower depth range of 8.5–10 km, with a peak slip of 5 m, and another in the depth range of 10.5–20 km, with a peak slip of 5.3 m. Both controlled the principal deformation signals in geodetic images. The slip was concentrated, along with a strike distance of 20 to 40 km, at a depth of 10 to 20 km. The earthquake was caused by the Zagros Mountains Front Fault (ZMFF), based on the results of 3D co-seismic deformation, inferred slip, preliminary investigation, and interpretation of the mainshock, as well as aftershock distributions.

## 1. Introduction

The Zagros orogenic belt is one of the salient convergent boundaries formed due to the Arabian-Eurasia continental collision [1]. The convergence between northeast-southwest striking faults represents right-lateral motion and corresponding shortening [2]. Shortening occurred in the Zagros belt at a rate of ~5–10 mm/year [3,4]. However, recent work investigating the north motion of the Arabian plate, relative to the Eurasia plate, is about 2 mm/year [5]. The Main Zagros Thrust, a young active right-lateral fault, bound the Zagros belt on the northeast [6]. The Arabian Plate’s northeastern boundary was formed by the Mid-Permian rifting and the Triassic break-up of Pangaea. As a result, a broad and shallow maritime shelf formed across the Mesozoic Arabian passive edge in the northeast [7]. The South Atlantic opened and extended, which was communicated onto the northeast of the Arabian Plate by accentuating subduction of the Neo-Tethys beneath Eurasia plate. This resulted in the Arabian Plate border dipping and gradually drifting, generating compression, and, finally, the obduction of the young Neo-Tethys ophiolitic oceanic crust onto the Arabian Plate’s northeast margin [8]. As reported in [9], the present-day tectonic configuration of the northeast part of the Arabian margin resulted from the final closure of the Neo-Tethys Ocean and continental collision between the Arabian Plate margin and Eurasia plate during the Early to Mid-Miocene. Progressive shortening led to the faulting, folding, and imbrication of the Arabian Plate margin sequences conveyed by the over-thrusting of the Tethyan accretionary prism components, which led to the final development of the Zagros Suture Zone (Figure 1). According to [10], this tectonic history is proved by the development of numerous major tectonic boundaries and zones associated along the Zagros tectonic strike. These boundaries are: (1) the Urumieh-Dokhtar Magmatic, (2) the Zagros Main Reverse Fault, (3) the magmatic Sanandaj-Sirjan zone, (4) the Zagros Thrust Front, (5) the Zagros Imbricate Zone, (6) the High Zagros Reverse Fault, (7) the Zagros Fold Zone, (8) the Zagros Foredeep Fault, and (9) the Mesopotamian Zone. Many authors have studied these boundaries and faults [5]. A regional geologic cross-section showing major tectonic divisions and major tectonic boundaries is shown in Figure 2. The DInSAR approach is one of the most extensively used methods for mapping surface deformation. This approach produces highly accurate spatially dense deformation fields [11]. Line of sight (LOS) measures are only sensitive in one dimension and cannot determine the total co-seismic surface deformation [12]. The MAI interferometry approach was recently used to detect the deformation along the azimuth direction. As a result, MAI technology was critical in relocating the measurement of 3D surface displacement [13].

According to the U.S. Geological Survey (USGS), an earthquake of a magnitude of 7.3 struck western Iran/eastern Iraq on 12 November 2017. The earthquake’s epicenter was located near the ZMFF at 45.9° E, 34.93° N, at a depth of 20 km. The USGS suggested the mainshock was mostly northwest-trending oblique-thrust faulting with shallow dipping to northeast at mid-crustal depth (19 km), according to focal mechanism determined from seismic waveform data. However, within a month of the Sarpol Zahab earthquake, aftershocks struck at depths of 7 to 12 km [14,15]. The distance between the aftershock cluster and the earthquake centroid has raised questions about (1) whether the rupture deployed on ground surface, and (2) how was the slip distributed due this event. Concerning the first object of this research, co-seismic deformation fields from ascending and descending (ALOS-2) Sentinel-1 images were obtained. Then, using DInSAR and MAI displacement measurements, we reconstructed the 3D co-seismic surface displacement field of the 2017 Sarpol Zahab earthquake. Finally, we characterized the potential of the seismogenic fault rupturing to the earth’s surface using 3D deformation fields. Furthermore, we inverted the DInSAR deformation data using the elastic dislocation to derive the fault shape and slip. This work detected a significant competency and poses a significant question: where was the slip concentrated on rupture?

## 2. Materials and Methods

The co-seismic surface displacements caused by the Sarpol Zahab event on 12 November 2017, were measured using two ascending and two descending interferograms from the Advanced Land Observing (ALOS-2) and Sentinel-1 sensors. The ALOS-2 sensor, in ascending and descending orbits, has a radar incidence angle of 47.2° with HH polarization and pixel spacing of 20 m in range and 25 m in azimuth. The range and azimuth of the visual scene are approximately 350 km by 350 km. The ascent and descent angles are 91° and 170°, respectively. The ALOS-2 has a longer wavelength of 23.6 cm (L band) and is less affected by SAR signal temporal decorrelation. We collected SAR images before and after the Sarpol mainshock to avoid the detrimental effect of the co-seismic signal on the recovered co-seismic DInSAR deformation field. The observed timing of the SAR images closest to the 2017 Sarpol Zahab earthquake was used in the DInSAR deformation field. The detailed information and ground coverage of the employed SAR pairs are depicted in Table 1 and Figure 1. The Sentinel-1A was acquired in Interferometric Wide (IW) mode on both ascending and descending orbits, with a radar incidence angle of 39.2° and VV polarization, with heading angles of 167°. The image scene covers approximately 170 km and 250 km, with pixel spacing of 25 m and 5m in range and azimuth, respectively. This type of image has a wavelength of 5.6 cm (C band). To remove the topographic phase component from the original interferogram, the Digital Elevation Model Mission v.4 (SRTM-4 DEM) was introduced [16]. The DInSAR approach was utilized to measure surface deformation.

Many scholars have described the basics of InSAR [17]. This method is based on the phase difference between two (or more) SAR images collected before and after an earthquake; a component of the phase difference corresponds to a one-dimensional measure of surface deformation in the satellite line-of-sight direction. They are known as interferograms, and the phase difference is in relation to the change in distance between the satellite and the image area. However, the phase cause phase shifts by atmospheric path delay (Δϕ_atm_), the flat earth effect, the contribution from surface deformation (Δϕ_def_), the topography (Δϕ_topo_), inaccuracies in the orbit state vectors (Δϕ_orb_), and other noise contributions (Δϕ_noise_). The difference in interferogram phase (ΔΦ) is measured in radians between 0 and 2π [18].
(1)$$\phi =2\pi k+\Delta \phi_{\mathrm{topo}}+\Delta \phi_{\mathrm{disp}}+\Delta \phi_{\mathrm{orb}\;}+\Delta \phi_{\mathrm{atm}\;}+\Delta \phi_{\mathrm{noise}}.$$

To find the surface deformation component (Δϕ_def_) in the target, in terms of line of sight (LOS) of the absolute phase difference in an interferogram (Δϕ), topographic contributions (Δϕ_topo_) are removed using a Digital Elevation Model (DEM). The inaccuracies come from the sensor and the data processing (ε). Suppose the atmosphere has similar spatial conditions between the two scan times. In this case, its contribution tends to be zero, and the distortion map can be obtained after removing the two-dimensional phase ambiguity in the interferometric model’s phase shift the remainder of the differential interferogram is shown in the expression below. To demonstrate the reliance of the perpendicular baseline B, we preserve the topographic phase (second term to the right of 1). We can use Equation (2) [18].
(2)$$\phi =\frac{4\pi B}{\lambda \;\mathrm{Rsin}\;\left(\theta \right)}+\frac{4\pi }{\lambda }\;\phi_{\mathrm{displ}}+\varepsilon .$$
where θ is the angle of incidence, R is the distance of change between the satellite sensor and the ground target, and ϕ_def_ is the deformation within the LOS component.

Multiple aperture interferometry is a technique that uses two SAR (Synthetic Aperture Radar) images to measure ground displacement along the satellite’s flight path (azimuth offset). The AMI is able to measure movement in the direction of flight by applying more elaborate signal processing. The main advantage of MAI is that two single-look-complex (SLC) SAR images with forward and backward looking are created by dividing the length of a synthetic aperture with signal processing. Standard InSAR processing is applied to forward- and backward-looking images, acquired at different periods, to produce a forward and backward SAR interferogram. Despite the fact that both of these interferograms indicate LOS displacement, the LOS directions are different. Subtracting the precision of the measurements is lower than that of regular DInSAR, sometimes decreasing to a few dozen centimeters [19]. It can be defined as follows:(3)$$\phi_{\mathrm{MAI}}=-4\pi /l*\mathrm{nx}.$$
where x is along-track displacement, l is adequate antenna length, and n is the faction factor of the full aperture.

The LOS component was obtained by a two-pass differential method with GAMMA software, as described by [20]. The data processing included the following procedures. The coherent coefficient methodology was applied for registration between the two images (master and slave). The process was accomplished by selecting an adequate and high consistency of points distributed far from the deformation region in the interferograms. The procedure was performed as explained in [17]. The phase was generated with a multi-look of 24 and 5 in azimuth and range, respectively.

In addition, phase noises were reduced on the differential interferograms by applying the Goldstein filters method. The topographic effect was removed by utilizing SRTM (DEM) with a resolution of 90 m in the phase. The removal was followed by the procedure provided in [21]. The minimum cost flow (MCF) method was implemented for unwrapping the phase. The deformation in the LOS direction was obtained after the geocoding process by geographic projection coordinate system (WGS84). In this work, both forward- and backward-look interferograms were individually produced by using SLC images. Multi-look processes with both 13 and 2 in azimuth and range were applied to enhance the MAI interferograms signal-to-noise ratio (SNR) during the operation. The Goldstein filtering method was utilized for forward-looking and backward-looking interferograms with a window size of 32. The process was followed by using a complex multiplication between the sub-aperture phases for ALOS-2 platform and Sentinel-1A data to generate the interferogram. Then, the flat-earth effects were removed from the phase by using the polynomial fitting method. Finally, the unwrapping phase was calculated, and a geocode was obtained using the coordinate system of a geographic projection (WGS84) provided by the authors of [22].

The three-dimensional co-seismic deformations were reconstructed by combining the measurements of LOS and along-track data. The imaging geometry of 3D vectors for surface deformation for the dot P in yellow color, the vector for DInSAR and MAI measurers (Figure 3) defined as Equations (4) and (5) [23].
(4)$${\mathrm{def}}_{\mathrm{los}}={\mathrm{def}}_{u\;}*\sin\left(\theta \right)+{\mathrm{def}}_{n\;}*\sin\left(\alpha \right)-{\mathrm{def}}_{e\;}*\cos\left(\alpha \right)*\sin\left(\theta \right).$$
(5)$${\mathrm{def}}_{\mathrm{AZ}}={\mathrm{def}}_{n\;}*\sin\left(\alpha \right)-{\mathrm{def}}_{e\;}*\cos\left(\alpha \right).$$
where θ represents the incidence angle of radar, α is the azimuth angle, def is the surface deformation observation, with sub-index, los and AZ are the observations of range and azimuth direction, respectively, and e and n are the east and north direction, respectively.

We restructured the 3D surface deformation map based on Equations (4) and (5). Surface deformation and the integration of InSAR and MAI data sets can be expressed as follows (6): (6)d=A∗u.
where d is a matrix containing our data, A is a design matrix, and u is the unknown 3D components.

Expanding Equation (6) using angles, as in Figure 2, Equations (7)–(9) can be generated:(7)$$u_{\mathrm{ew}}\;=\frac{{\mathrm{def}}_{\mathrm{des}}^{\mathrm{LOS}}}{\sin\left(\theta_{\mathrm{desc}}\right)\cos\left(\alpha_{\mathrm{desc}}\right)}-\frac{{\mathrm{def}}_{\mathrm{asc}}^{\mathrm{LOS}}}{\sin\left(\theta_{\mathrm{asc}}\right)\cos\left(\alpha_{\mathrm{asc}}\right)}+\frac{{\mathrm{def}}_{\mathrm{asc}}^{\mathrm{azi}}}{\sin\left(\alpha_{\mathrm{asc}}\right)}+\frac{{\mathrm{def}}_{\mathrm{des}}^{\mathrm{rng}}}{\sin\left(\alpha_{\mathrm{des}}\right)}.$$
(8)$$u_{\mathrm{ns}}\;=\frac{{\mathrm{def}}_{\mathrm{des}}^{\mathrm{LOS}}}{\sin\left(\theta_{\mathrm{desc}}\right)\sin\left(\alpha_{\mathrm{desc}}\right)}-\frac{{\mathrm{def}}_{\mathrm{asc}}^{\mathrm{LOS}}}{\sin\left(\theta_{\mathrm{asc}}\right)\sin\left(\alpha_{\mathrm{asc}}\right)}+\frac{{\mathrm{def}}_{\mathrm{asc}}^{\mathrm{azi}}}{\cos\left(\alpha_{\mathrm{asc}}\right)}-\frac{{\mathrm{def}}_{\mathrm{des}}^{\mathrm{rng}}}{\sin\left(\alpha_{\mathrm{des}}\right)}.$$
(9)$$u_{\mathrm{ud}}\;=\frac{{\mathrm{def}}_{\mathrm{des}}^{\mathrm{LOS}}}{\cos\left(\theta_{\mathrm{desc}}\right)}+\frac{{\mathrm{def}}_{\mathrm{asc}}^{\mathrm{LOS}}}{\cos\left(\theta_{\mathrm{asc}}\right)}\;.$$
where def is deformation observations, sub-indices (asc) and (desc) represent ascending and descending orbit passes, respectively; super-indices represent is phase measurements (LOS), azimuth (azi), and range(rng) offsets, respectively; and factor u is the estimated unknown component of the displacement in east, north, and up.

**Figure 3 ijerph-19-09831-f003:**
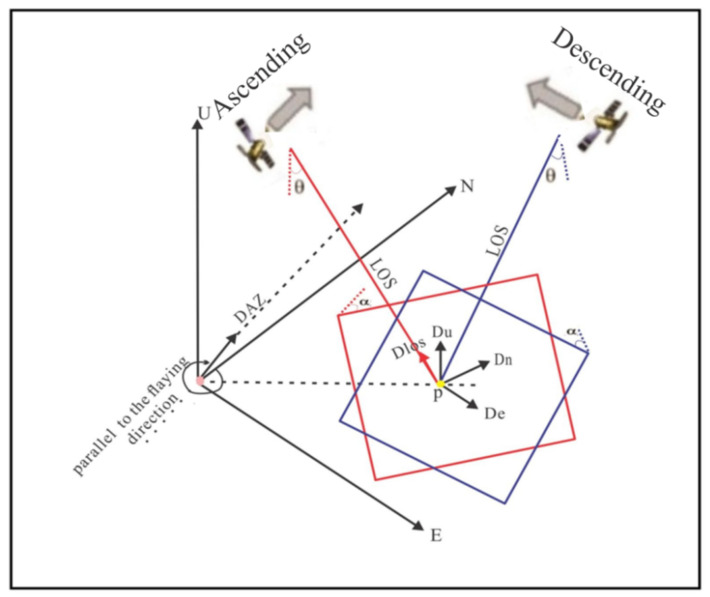
Geometrical projection relationships between the SAR imaging geometry and the three-dimensional (3D) motion components, up-down deformation (Du), north-south deformation (Dn), and east-west deformation (De), above a specific area (P), with a SAR sensor having a right lateral look (red ascending and blue descending). DAZ correspond on deformation along azimuth, direction incidence angle of radar (θ) and azimuth angle (α).

## 3. Results

### 3.1. Co-Seismic Deformation along the Line-of-Sight

The co-seismic displacement was obtained by the DInSAR method described in Section 2 for ALOS-2 and Sentinel-1 sensor in both tracks (ascending and descending). Two lobes distinguish the deformation with a distinctive elliptical shape. The first lobe is located roughly southwest of the quake’s epicenter in both ascending and descending data. This lobe has substantial positive LOS displacement, and the horizontal movement indicates that this zone of the ground experienced uplift. The maximum displacement on the LOS was 100 and 50 cm (Figure 4 and Figure 5, respectively). The second lobe exists roughly northeast of an epicenter of a quake with the maximum displacement on the LOS of −15 and −35 cm (Figure 4) and (Figure 5) respectively. The descending data show more considerable surface displacements than that of the ascending DInSAR data. The trace fault is oriented to northwest southeast. The co-seismic ground displacement data obtained by SAR show a large co-seismic slip, defined mostly by the strike-slip component that is compatible with the Zagros Mountains Front Fault. The fringe density of the interferogram of ALOS-2 data (L-band) is lower density than the Sentinel-1 data (C-band), though the second one contains noise.

### 3.2. Along-Track Co-Seismic Deformation (Azimuth)

The InSAR measurement is sensitive and more evident in the LOS deformation than the azimuth displacement (Figure 4 and Figure 5). In contrast, the MAI method is not affected by the atmospheric signal delay. Hence, it may be more authoritative than InSAR in specific places where atmospheric effects are severe [24]. The ascending of ALOS-2 and descending of Sentinel-1A showing opposite direction of the movement on the right-hand side of the fault with a maximum value of 100 cm (Figure 6a,b). The displacement pattern is consistent with a component of strike-slip motion. The InSAR appears to have low sensitivity to azimuth deformation, while the MAI is useful to measure the deformation along the azimuth.

### 3.3. 3D Co-Seismic Deformation

We retrieved the 3D displacement by integrating the measurements of the LOS and along-track from the ALOS-2 and Sentinel-1A data. The east, north, and up components were retrieved based on the 3D displacement field method, described in Section 2. Figure 7 shows the calculated three-dimensional displacement field. The east component showed an asymmetric pattern of deformation. The motion was more in the western part of the surface rupture than the eastern (Figure 7a,d). The maximum relative surface displacement on the sides of the fault was −50 cm. In the up component (Figure 7b,e), the maximum surface displacement was 100 cm. This component showed similar deformation patterns with the LOS displacement measured by DInSAR (Figure 4d and Figure 5d). The north component showed patterns almost parallel to the surface rupture. The large signal of peak deformation occurred at approximately 0.5 km on either side of the fault, suggesting that the earthquake involved predominantly strike-slip motion rupture (Figure 7c,f). The maximum relative surface displacement was 100 cm.

### 3.4. Co-Seismic Faults and Slip Inversion

Recently, LOS deformation maps have become a source of beneficial information for elicitation of the slip along the fault plane [25]. Inversion of the geodetic modeling is required to estimate the fault and slip distribution along the Sarpol Zahab rupture. We utilized the rectangular dislocation in a homogeneous elastic half-space [26] to solve faults and slip using the SAR data described in Section 2. We primarily reduced the observed data, without losing important information, by using a two-dimensional quantization algorithm [25]. We maintained sampling in the co-seismic zone for ALOS ascending and Sentinel-1 descending track. We carried out the inferred source model by applying the nonlinear algorithm with a larger 5 km × 5 km size. Then, a simulated annealing algorithm, with a smaller size of 2 km × 2 km, was derived to find the optimized fault parameter [27]. We found three sub-fault segments from the flat-ramp fault geometry parameters, including strike angle and the up-dip depth. We fixed the location, length, and dip based on the preliminary USGS solution for each of these segments. The best-fit parameter is summarized in Table 2. Laplace smoothing was used for rectangular fault orientation at approximately 337.5° NW, and the dip angle was 11.2° (Figure 8a). Our result suggests that the fault rupture spread from the Northeast to the Southwest. The seismic moment was freed at a depth of 8.5–10 km and 10.5–20 km. The slip occurred Southwest of the epicenter, as provided by the USGS. The maximum slip was concentrated, along with a strike distance of 20–40, at a depth of 10–14 km (Figure 8b). The total geodetic moment calculated was 1.15 × 1020 Nm, equivalent to a Mw 7.31 earthquake. The obtained geodetic moment was consistent with the seismic moment identified by the USGS. The amount of slip gently wastes away to zero northeast of the area of maximum slip. The final model for InSAR predictions resembled the data well (Figure 9). The residual was large signals identified from the north-south components (Figure 9f,l). The east-west and up-down components were distinguished by less residual signals (Figure 9d,e,j,k).

## 4. Discussion

Due to DInSAR’s limited sensitivity to azimuth deformation, all of the data show significant discrepancies between the various SAR sensors and orbits. Moreover, the strike of the co-seismic motions is parallel to the ascending orbit and crosses the descending orbit, and the fault is confirmed as a thrust fault; consequently, variations in the projected values are unavoidable. Meanwhile, because the fringe density in larger magnitude earthquakes is too great to effectively conduct phase unwrapping, the L-band ALOS-2 SAR sensor has reliability in such severe deformation monitoring. The MAI approach, based on phase differencing, performs better. From our results, we confirmed that the UD components are the most violent, and westward motion has a significant influence on horizontal motion (Figure 8b). The geodetic data revealed two substantial slip sources: one at a shallower depth of 8.5 to10 km, with a peak slip of 5 m, and another in the depth range of 10.5–20 km, with a peak slip of 5.3 m. Both are in control of the major deformation signals in geodetic images. The co-seismic inversion result of the 2017 Sarpol Zahab earthquake revealed that the rupture depth ranged between 10 km and 20 km. These ranges are similar to those of the historical earthquakes (Mw > 6) in Zagros mentioned by Huang et al., in 2019. Additionally, the focal mechanism solutions from the Global Centroid Moment Tensor (GCMT) and the U.S. Geological Survey (USGS) suggest that the Sarpol Zahab earthquake occurred at a centroid depth of 20 km. It provides the fault plane parameters of the Sarpol Zahab earthquake, inverted from seismology and line-of-sight displacements. The results confirmed that the northwest strike, in the general vicinity of this quake, drove the Zagros Mountains uplift. The northeast dip of the co-seismic fault is consistent with the rupture related to the ZMFF.

## 5. Conclusions

We inferred the 3D surface deformation of the Sarpol Zahab earthquake by combining DInSAR and MAI interferograms from co-seismic ALOS-2 and Sentinel-1A data. The observed surface displacements in the east, up, and north directions were approximately 50, 100, and 100 cm, respectively. The best-fit faulting geometry had a strike trend of 337.5°, with a dip of 11.2°.

The total seismic moment was 1.151020 nm, corresponding to Mw 7.31. The dip-depth ranges were 8.5–10-km and 10.5–20 km, with a peak slip of 5 m and 5.3 m, respectively, and concentrated along a 20–40 km strike distance at a depth of 10–14 km. When compared to prior studies’ results, the 3D-based model yielded more precise parameter estimations, with significantly reduced uncertainty.

## Figures and Tables

**Figure 1 ijerph-19-09831-f001:**
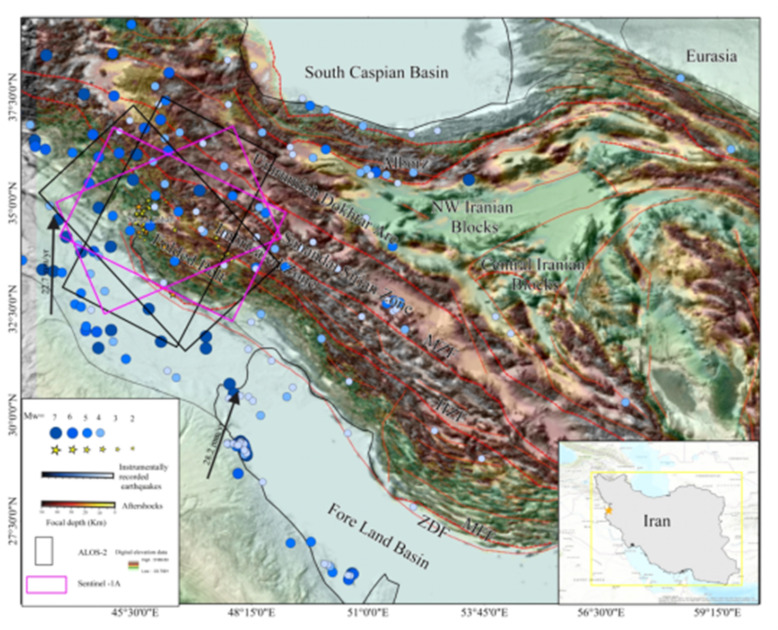
Location map of the study area with tectonic, topography, and major faults of The Zagros Mountain. The black arrows represent the Arabian plate GPS velocity relative to the stable Eurasian plate and the GPS velocity field, respectively [5]. The aftershocks are represented by color ramp with magnitude value 7-2. Instrumentally recorded earthquakes are distinguished by color ramp with magnitude 7-4, and the epicenter of the Mw 7.3 is represented by an orange star. The main strike fault is depicted in black. The red lines represent the potential locations of “master blind thrusts”, which are the High Zagros Fault (HZF), the Main Recent Fault (MRF), and the Zagros Foredeep Fault (MFF). The black circle represents the cities surrounding the earthquake. The spatial coverage of the ALOS-2 data is depicted by black rectangles, and Sentinel-1A data is depicted by purple rectangles.

**Figure 2 ijerph-19-09831-f002:**
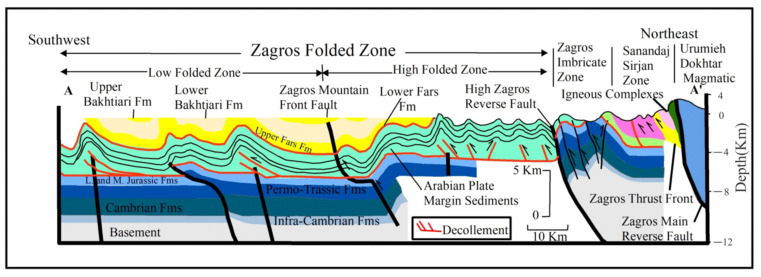
Regional geologic cross-sections across northeast Iraq showing major tectonic divisions and major tectonic boundaries (Modified after [10]).

**Figure 4 ijerph-19-09831-f004:**
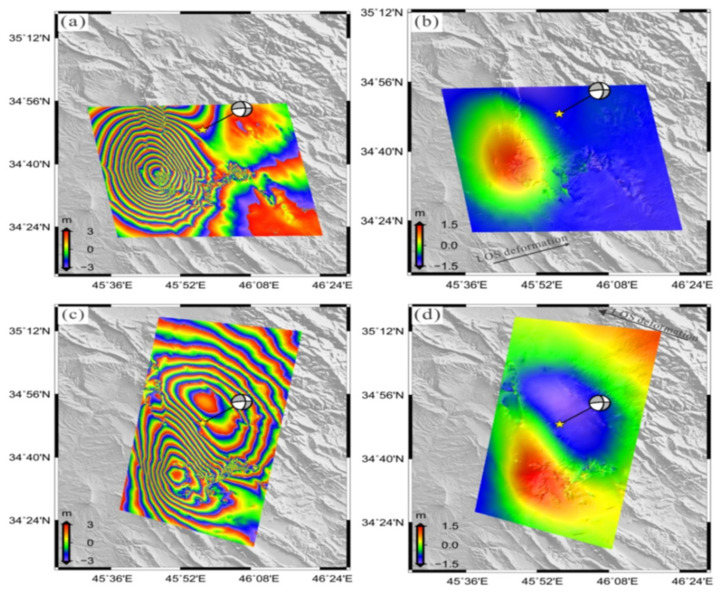
The observed co-seismic deformation of ALOS-2 SAR images ((**a**,**b**) ascending orbit, (**c**,**d**) descending orbit). The left-hand image shows differential interferogram, and the right-hand image shows LOS co-seismic deformations. The colorized images of wrapping intervals are 0.03 cm along LOS. The yellow star represents the epicenter of the mainshock.

**Figure 5 ijerph-19-09831-f005:**
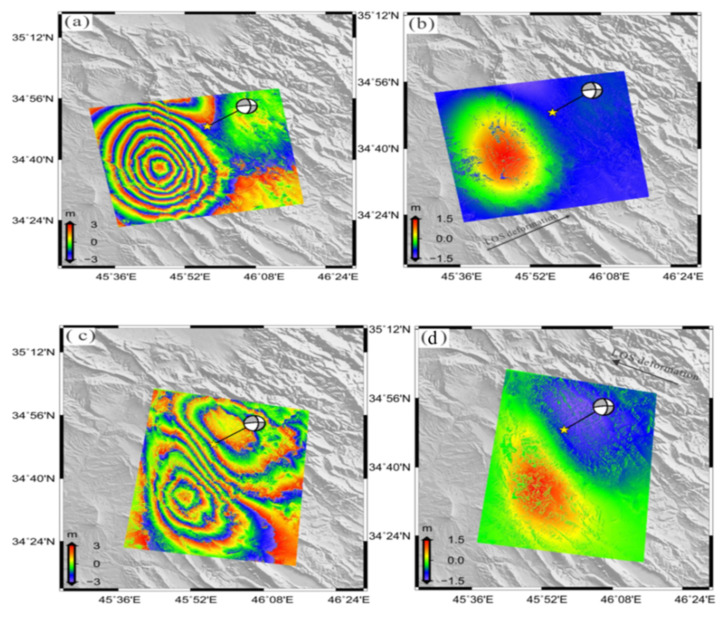
The observed co-seismic deformation of Sentinel-1A images ((**a**,**b**) ascending orbit, (**c**,**d**) descending orbit). The left-hand image shows differential interferogram, and the right-hand image shows LOS co-seismic deformations. The colorized images of wrapping intervals are 0.03 cm along LOS. The yellow star represents the epicenter of the mainshock.

**Figure 6 ijerph-19-09831-f006:**
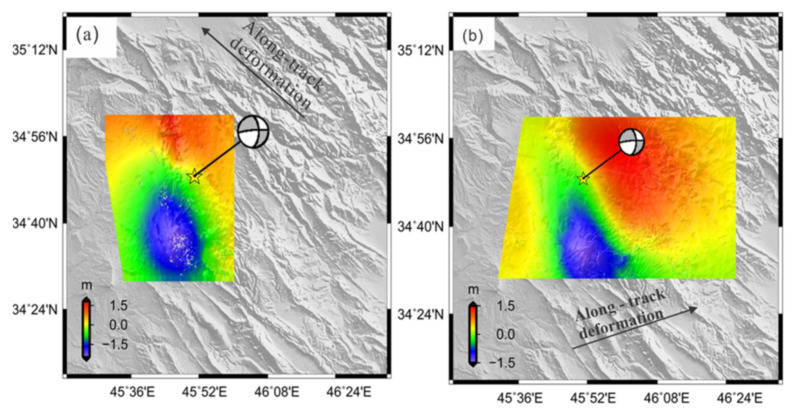
Observed co-seismic deformation along azimuth direction: (**a**) ALOS-2 image (**b**) Sentinel-1 images. The yellow star represents the epicenter of the mainshock.

**Figure 7 ijerph-19-09831-f007:**
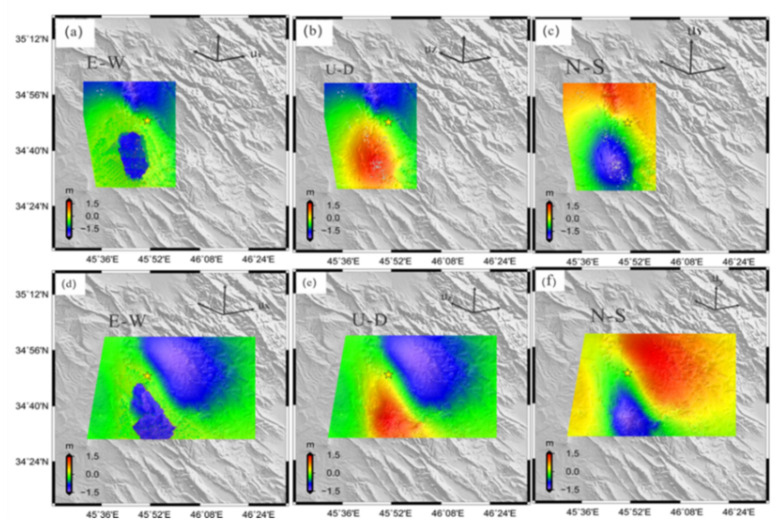
(**a**–**c**) Show the 3D surface displacement maps retrieved from integrating LOS and MAI measurements of ALOS-2 SAR images, and (**d**–**f**) show the Sentinel-1 SAR images; (**a**,**d**) east-west component, (**b**,**e**) up-down component, and (**c**,**f**) the north-south component. The yellow star represents the epicenter of the mainshock.

**Figure 8 ijerph-19-09831-f008:**
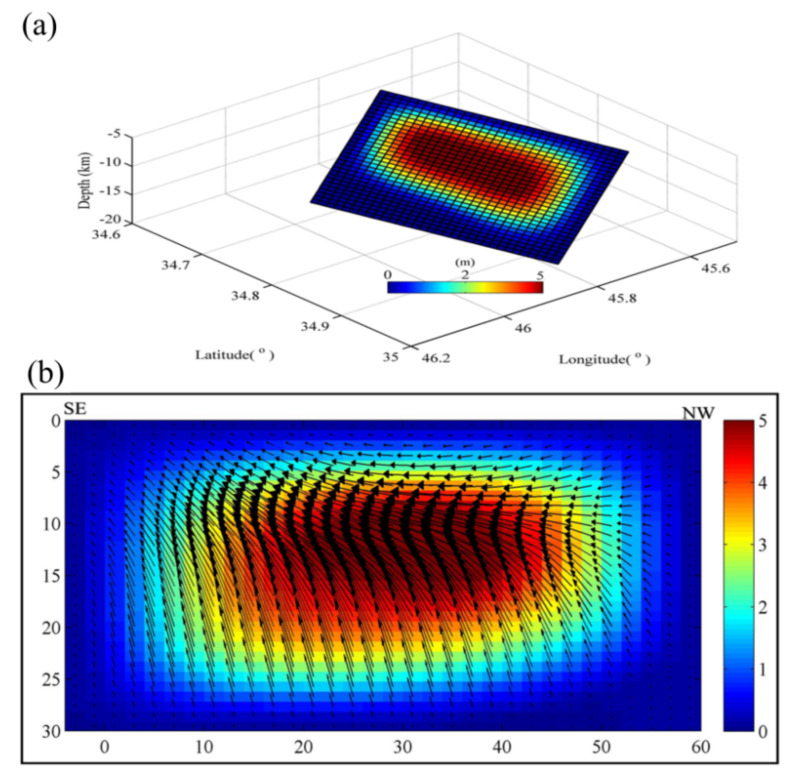
(**a**) Inferred Fault (**b**) Slip distribution.

**Figure 9 ijerph-19-09831-f009:**
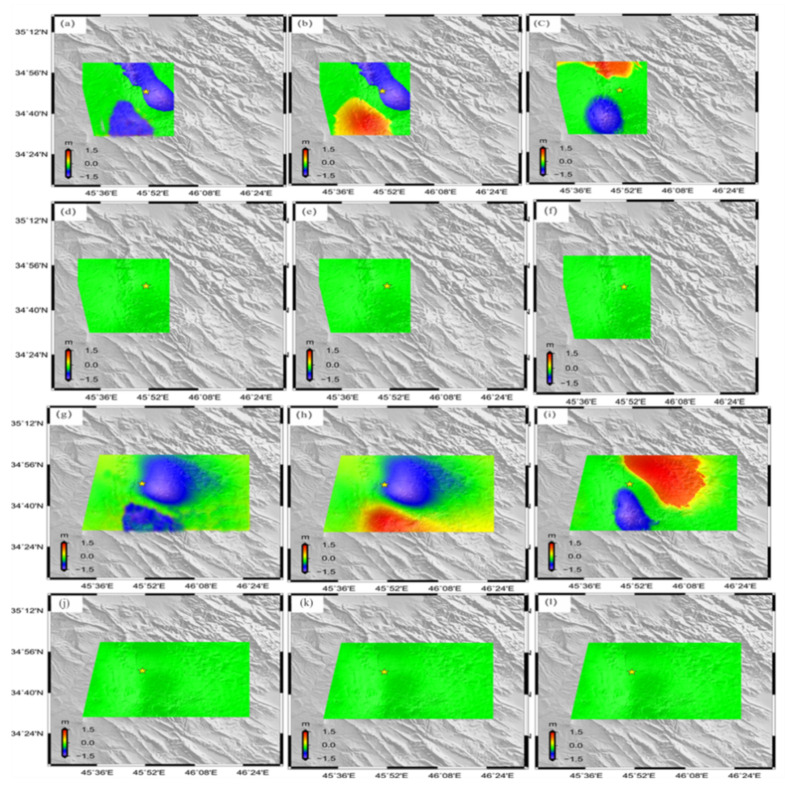
Model prediction and residuals for 3D observations the left-hand image shows the east-west component, medial image shows the up-down component and the right image shows the north-south component. 3D ALOS-2 model prediction (**a**–**c**) and the residual (**d**–**f**). 3D Sentinel-1 descending model prediction (**g**–**i**) and the residual (**j**–**l**). The yellow star represents the epicenter of the mainshock.

**Table 1 ijerph-19-09831-t001:** Dataset of Sarpol Zahab earthquake 2017.

Sensor	Orbit Path	Acquisition Data	Incidence Angle	Perpendicular Baseline	Heading Angle (°)
ALOS-2	ascending	9August 2016 14 November 2017	47.2	70 m	91
	Descending	4 October 2017 15 November 2017	47.2	186.3 m	170
Sentinel-1A	ascending	11 November 2017 23 November 2017	39.2	62.2m	167
	descending	7 November 2017 19 November 2017	39.2	70.2	167

**Table 2 ijerph-19-09831-t002:** The best-fit faulting parameters of the 2017 Sarpol Zahab earthquake, inverted from seismology and line-of-sight displacements.

Parameter	Strike (◦)	Dip (◦)	Rake (◦)	Length (km)	Width (km)	Depth (km)	Slip (m)	Longitude (◦)	Latitude (◦)
USGS ^1^	352	16	137	1.5	-	−19	3.26	45.956	34.905
Global MT ^2^	351	11	140	-	-	18.0	-	45.84	34.83
Barnhart et al., (2018) ^3^ [28]	350	15	128 ^a^	-	-	15.0	-	45.87 ^b^	34.65 ^b^
Feng et al., (2018) ^3^ [29]	353.5	14.5	135.6 ^a^	-	-	14.5	-	45.86 ^b^	34.73 ^b^
This study (InSAR) ^3^	337.5	11.2	130.1 ^a^	30	30	-8.5	5	45.6 ^b^	34.6 ^b^
	337.5	11.2	135 ^a^	30	30	10.5	5.3	45.7 ^b^	34.8 ^b^

^1^ USGS United States Geological Survey; ^2^ Global CMT stands for Global Centroid Moment Tensor Project; and ^3^ The various model fault planes derived from co-seismic interferograms’ elliptical fringe pattern; ^a^ Each fault patch determines the mean rake direction; ^b^ The longitude, latitude, and depth are defined as the fault plane’s centroid.

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
