# Peer review of "3D Co-Seismic Surface Displacements Measured by DInSAR and MAI of the 2017 Sarpol Zahab Earthquake, Mw7.3"

_ijerph, 2022, doi:10.3390/ijerph19169831_

Round 1

Reviewer 1 Report

Tha topic of the article is well chosen, being an interesting subject. The introduction can be improved with more relevant references, because there exists a wealth of scientific literature on this approach. 

The workflow is described according to the followed steps in processing. 

The main shortcoming remain the discussion section and the conclusion area, which are very poorly built. 

Author Response

Response to Reviewer 1 Comments

Dear Editor and dear Reviewers

Thank you very much for your comments and suggestions. We have modified the

manuscript according to the comments below.

The revised paper has marginal improvements over the previous submitted version. Based on my 1st review, the authors responses to reviewer and the revised version there are still few major points that should be clarified and addressed (see major comments to authors from my 1st review). Furthermore, my major comments are not fully addressed in the revised version (see also authors responses to reviewer).

Major Comments:

Point 1:  The main shortcomings remain the discussion section and the conclusion area, which are very poorly built. 

Response 1:

  • Thanks so much, regarding this point, we improved discussion section and the conclusion area and it is clearer now Line 332-353.

On behalf of all the co-authors

Yours sincerely

Randa G. Ali

Reviewer 2 Report

This paper looks at monitoring using the latest technology in the field, it considers recent advances in research and adopts an ambitious and risky approach to investigate the topic. Seismic displacements are of utmost importance and evaluation of them is of the highest priority. This paper brings a novel insight into this problem. 

This is amazing work and it has the potential to have great impact.

I suggest acceptance.

Author Response

Response to Reviewer 1 Comments

Dear Editor and dear Reviewers

This paper looks at monitoring using the latest technology in the field, it considers recent advances in research and adopts an ambitious and risky approach to investigate the topic. Seismic displacements are of utmost importance and evaluation of them is of the highest priority. This paper brings a novel insight into this problem. 

This is amazing work and it has the potential to have great impact.

Thank you very much for your positive comments, and we really appreciate your words

On behalf of all the co-authors

Yours sincerely

Randa G. Ali

Reviewer 3 Report

By targeting 3D co-seismic surface displacement of the 2017 Sarpol Zahab Mw 7.3 earthquake, this paper combined DInSAR and MAI interferograms of co-seismic of ALOS-2 and Sentinel -1A 347 data and estimated surface displacement in the east, up, and north directions at 50, 100, and 100 cm, respectively, with the best-fit fault strike of 337.5° and a dip of 11.2°. this paper also estimated the total seismic moment of 1.151020 nm, corresponding to Mw 7.31, and the maximum slip of ~5m. The co-seismic NE fault dipping is consistent with the Zagros Mountains frontal fault. The conclusions were supported by the data presented in the paper.

This paper was good organized and the figures are clear and easy to follow. However, there are numbers of spelling and writing problems. This paper need essential writing polishing before accept for publication in IJERPH.

General comments are listed, but not limit as follows:

Line 17, 100, 50 and 100 cm? These are very small displacements!

Line 28, organic--orogenic

Lines 29-30, this sentence doesn’t present clear meaning

Line 31, why do you use “however”? the rate of 2 mm/year is logically less than the Zagros belt of 5-10 mm.

Line 33, (Figure 1)

Line 39, here and throughout the manuscript, consisting using “Eurasia plate” to replace Eurasia  

Line 42, margin's northeast--northeast margin?

Line 44, according to [9] is not a standard usage.

Line 44, imbrication consists of faulting and folding.

Line 47, .Figure2.----(Figure 2), similar problems throughout the manuscript.

Lines 47-48, not a complete sentence—Figure 2? According to [10]—very confused presentation, and how a tectonic history proved by XXX boundaries?

Lines 53-54, not a complete sentence, and [5] represents authors?? Or literatures??

Line 58, what is azimuth deformation?

Line 65 genital?? There are so many spell problems and grammar mistakes throughout the manuscript. This paper is very hard to follow  

Lines 68-69, whether is the rupture located on ground surface? Why have such a question?

Lines 76-77, where was the slip concentration on rupture?

Line 110, equation (1), also you need define variant

Line 191, 3.1.---3.1

Line 207, 3.2.---3.2

Line 217, 3.3.3. D----3.3 3D co-seismic deformation

Line 223, and throughout the manuscript, cite a Figure SHOULD follow the format prescribed in literature, but rather than this following the end of a sentence.

Line 336, the new result of rupture depth of 20-40 km has too broad range to determine.

Authors need check and do essential revisions throughout the manuscript.

Author Response

Response to Reviewer 1 Comments

Dear Editor and dear Reviewers

Thank you very much for your comments and suggestions. We have modified the

manuscript according to the comments below.

The revised paper has marginal improvements over the previous submitted version. Based on my 1st review, the authors responses to reviewer and the revised version there are still few major points that should be clarified and addressed (see major comments to authors from my 1st review). Furthermore, my major comments are not fully addressed in the revised version (see also authors responses to reviewer).

Major Comments:

Point 1:  100, 50 and 100 cm? These are very small displacements!

Response 1:

  • Thanks so much, regarding this point, in previous version we inserted unmodified unite, now we have updated the unit to meter and it is clearer now Line 17.

Point 2: organic--orogenic.

Response 2:

  • Thank you, the word have been modified Line 30.

Point 3: Lines 29-30, this sentence doesn’t present clear meaning.

Response 3:

  • The sentence have been modified now become clear meaning Line 31-32.

Point 4: Line 31, why do you use “however”? the rate of 2 mm/year is logically less than the Zagros belt of 5-10 mm.

Response 4:

  • Because the recent study have new rate of motion, the paragraph have been modified Line 34-35.

Point 5: Line 33, (Figure 1)

Response 5:

  • The pointhave been provided Line 51.

Point 6: Line 39, here and throughout the manuscript, consisting using “Eurasia plate” to replace Eurasia 

Response6

  • Thank you, the point have been provided.

Point 7: Line 42,margin's northeast--northeast margin?:

Response7:

  • The detailed have been modified Line 44-49

Point 8: Line 44, according to [9] is not a standard usage.

Response8:

  • The point have been modified Line 44-.

Point 9:

Line 44, imbrication consists of faulting and folding

Response9:

  • The point have been modified

Point 10:

Line 47, .Figure2.----(Figure 2), similar problems throughout the manuscript.

Response10:

  • This point have been provided Line 56.

Point 11:

Lines 47-48, not a complete sentence—Figure 2? According to [10]—very confused presentation, and how a tectonic history proved by XXX boundaries?

Response11:

  • This point have been modified Line 51.

Point12:

Lines 53-54, not a complete sentence, and [5] represents authors?? Or literatures??

Response12

  • This point have been modified Line 55- 56.

Point13:

Line 58, what is azimuth deformation?

Response13:

  • This point have been provided Line 62.

Point14:

Line 65 genital?? There are so many spell problems and grammar mistakes throughout the manuscript. This paper is very hard to follow  

Response14:

  • This point have been modified Line 67-69.

Point15:

Lines 68-69, whether is the rupture located on ground surface? Why have such a question?

Response15:

  • This point have been modified Line 51.

Point16:

Lines 76-77, where was the slip concentration on rupture?

Response16:

  • This point have been modified.

Point17:

Line 110, equation (1), also you need define variant ∅

Response17:

  • The InSAR phase ∅is a combination of several contributions (curved Earth, topography, surface displacements, atmospheric delays and phase noise).

Point18:

Line 191, 3.1.---3.1

Response18:

  • This point have been Provided Line 196.

Point19:

Line 207, 3.2.---3.2

Response19:

  • This point have been Provided Line 211.

Point20:

Line 217, 3.3.3. D----3.3 3D co-seismic deformation

Response20:

  • This point have been Provided Line 221.

Point 21:

Line 223, and throughout the manuscript, cite a Figure SHOULD follow the format prescribed in literature, but rather than this following the end of a sentence.

Response 21:

  • This point have been Provided.

Point 22:

Line 336, the new result of rupture depth of 20-40 km has too broad range to determine.

Response 22:

  • This point have been modified Line 345.

On behalf of all the co-authors

Yours sincerely

Randa G. Ali

Reviewer 4 Report

The reviewed article is very interesting. The article has a good research methodology, mathematical basis, and appropriate graphic appendices. If there is a record from the seismic stations of the earthquake under study, then I recommend that a graphic appendix with a record of the earthquake be added. The article, following this reviewer's recommendation, should be published.

Author Response

Response to Reviewer 1 Comments

Dear Editor and dear Reviewers

Thank you very much for your comments and suggestions. We have modified the

manuscript according to the comments below.

The revised paper has marginal improvements over the previous submitted version. Based on my 1st review, the authors responses to reviewer and the revised version there are still few major points that should be clarified and addressed (see major comments to authors from my 1st review). Furthermore, my major comments are not fully addressed in the revised version (see also authors responses to reviewer).

Major Comments:

Point 1:  The reviewed article is very interesting. The article has a good research methodology, mathematical basis, and appropriate graphic appendices. If there is a record from the seismic stations of the earthquake under study, then I recommend that a graphic appendix with a record of the earthquake be added. The article, following this reviewer's recommendation, should be published.

Response 1:

  • Thank you so much for your positive comments. As we used satellite data in this study, we think that the ground station record is not necessary in this work. However, in our future work we will consider this point. Once again thanks so much.

On behalf of all the co-authors

Yours sincerely

Randa G. Ali

This manuscript is a resubmission of an earlier submission. The following is a list of the peer review reports and author responses from that submission.